# Overexpression of MRP1/ABCC1, Survivin and BCRP/ABCC2 Predicts the Resistance of Diffuse Large B-Cell Lymphoma to R-CHOP Treatment

**DOI:** 10.3390/cancers15164106

**Published:** 2023-08-15

**Authors:** Danijela Mandić, Lana Nežić, Ljiljana Amdžić, Nataša Vojinović, Radoslav Gajanin, Miroslav Popović, Jugoslav Đeri, Milena Todorović Balint, Jelena Dumanović, Zoran Milovanović, Jelica Grujić-Milanović, Ranko Škrbić, Vesna Jaćević

**Affiliations:** 1Department of Hematology, Clinic of Internal Medicine, University Clinical Center Republic of Srpska, 12 Beba, 78000 Banja Luka, Bosnia and Herzegovina; danijela.mandic@kc-bl.com; 2Department of Internal Medicine, Faculty of Medicine, University of Banja Luka, Save Mrkalja 14, 78000 Banja Luka, Bosnia and Herzegovina; 3Department of Pharmacology, Toxicology and Clinical Pharmacology, Faculty of Medicine, University of Banja Luka, Save Mrkalja 14, 78000 Banja Luka, Bosnia and Herzegovina; lana.nezic@med.unibl.org (L.N.); ranko.skrbic@med.unibl.org (R.Š.); 4Center for Biomedical Research, Faculty of Medicine, University of Banja Luka, Save Mrkalja 14, 78000 Banja Luka, Bosnia and Herzegovina; ljiljana.amidzic@med.unibl.org (L.A.); natasa.vojinovic@med.unibl.org (N.V.); 5Department of Pathology, Faculty of Medicine, University of Banja Luka, Save Mrkalja 14, 78000 Banja Luka, Bosnia and Herzegovina; radoslav.gajanin@med.unibl.org; 6Department of Gynecology and Obstetrics, Faculty of Medicine, University of Banja Luka, Save Mrkalja 14, 78000 Banja Luka, Bosnia and Herzegovina; drpopovic.gin1@gmail.com; 7Department of Surgery, Faculty of Medicine, University of Banja Luka, Save Mrkalja 14, 78000 Banja Luka, Bosnia and Herzegovina; djeri@blic.net; 8Department of Hematology, Faculty of Medicine, University of Belgrade, 11000 Belgrade, Serbia; bb.lena@gmail.com; 9Clinic of Hematology, University Clinical Center of Serbia, 2 Pasterova, 11000 Belgrade, Serbia; 10Department of Analytical Chemistry, Faculty of Chemistry, University of Belgrade, Studenski trg 16, 11000 Belgrade, Serbia; dumanovicjelena@gmail.com; 11Medical Faculty of the Military Medical Academy, University of Defence, Crnotravska 17, 11000 Belgrade, Serbia; 12Special Police Unit, Ministry of Interior, Trebevićka 12/A, 11030 Belgrade, Serbia; tinahoks41@gmail.com; 13Institute for Medical Research, National Institute of the Republic of Serbia, Department for Cardiovascular Research, University of Belgrade, Dr. Subotića 4, 11000 Belgrade, Serbia; jeca@imi.bg.ac.rs; 14Department for Experimental Toxicology and Pharmacology, National Poison Control Centre, Military Medical Academy, Crnotravska 17, 11000 Belgrade, Serbia; 15Department of Chemistry, Faculty of Science, University of Hradec Kralove, Rokitanskeho 62, 50003 Hradec Kralove, Czech Republic

**Keywords:** multidrug resistance, non-Hodgkin lymphoma, MRP1/ABCC1, BCRP/ABCC2, survivin

## Abstract

**Simple Summary:**

This study aimed to investigate if the coexpression of ABC transporters and survivin is associated with R-CHOP treatment response. Bcl-2 was in strong positive correlation with clinical parameters and all biomarkers except P-gp/ABCB1. The overexpression of MRP1/ABCC1, survivin, and BCRP/ABCC2 presented as high immunoreactive scores (IRSs) was detected in Refractory and Relapsed groups, respectively, whereas the IRS of P-gp/ABCB1 was low. Significant correlations were found among either MRP1/ABCC1 and survivin or BCRP/ABCC2 expression in the Refractory and Relapsed groups, respectively. In multiple linear regression analysis, ECOG status along with MRP1/ABCC1 or survivin, and BRCP/ABCG2 was significantly associated with the prediction of the R-CHOP treatment response. DLBCL might have increased levels of certain molecules such as MRP1/ABCC1, survivin, and BCRP/ABCC2 that can predict resistance to R-CHOP.

**Abstract:**

Background: Approximately 40% of patients with diffuse large B-cell lymphoma (DLBCL) experience treatment resistance to the first-line R-CHOP regimen. ATP binding cassette (ABC) transporters and survivin might play a role in multidrug resistance (MDR) in various tumors. The aim was to investigate if the coexpression of ABC transporters and survivin was associated with R-CHOP treatment response. Methods: The expression of Bcl-2, survivin, P-glycoprotein/ABCB1, MRP1/ABCC1, and BCRP/ABCC2 was analyzed using immunohistochemistry in tumor specimens obtained from patients with DLBCL, and classified according to the treatment response as Remission, Relapsed, and (primary) Refractory groups. All patients received R-CHOP or equivalent treatment. Results: Bcl-2 was in strong positive correlation with clinical parameters and all biomarkers except P-gp/ABCB1. The overexpression of MRP1/ABCC1, survivin, and BCRP/ABCC2 presented as high immunoreactive scores (IRSs) was detected in the Refractory and Relapsed groups (*p* < 0.05 vs. Remission), respectively, whereas the IRS of P-gp/ABCB1 was low. Significant correlations were found among either MRP1/ABCC1 and survivin or BCRP/ABCC2 in the Refractory and Relapsed groups, respectively. In multiple linear regression analysis, ECOG status along with MRP1/ABCC1 or survivin and BRCP/ABCG2 was significantly associated with the prediction of the R-CHOP treatment response. Conclusions: DLBCL might harbor certain molecular signatures such as MRP1/ABCC1, survivin, and BCRP/ABCC2 overexpression that can predict resistance to R-CHOP.

## 1. Introduction

Diffuse large B-cell lymphoma (DLBCL) is the most common type of aggressive non-Hodgkin lymphoma. It shows a remarkable clinical and biological heterogeneity that can be explained by the existence of molecularly diverse subtypes [1,2]. By applying cell-of-origin (COO) classification, early gene expression profiling studies classified 80–85% of pretreatment DLBCLs into two subgroups, namely the activated B-cell (ABC) and germinal center B-cell (GCB), with the remaining cases termed “unclassified” which have been further defined into several genetic subtypes [1,2,3,4]. The combination of rituximab (R; an anti-CD20 monoclonal antibody) with the anthracycline-based chemotherapeutic regimen CHOP (cyclophosphamide, doxorubicin, vincristine, and prednisone) presents a core treatment of DLCBL that has substantially improved patient outcomes with prolonged disease-free survival rates [5,6,7]. However, up to 40% of all patients develop relapsed/refractory disease (R/R) where ABC-DLBCL generally shows a poorer prognosis and a higher incidence of R-CHOP resistance [1,7,8,9,10]. The prognosis of R-CHOP-resistant R/R DLBCL is dismal even after being subsequently treated with salvage chemotherapy and consolidative autologous stem cell transplantation [9,10].

Recognizing the biological heterogeneity and the genetic subtypes of DLBCL, it becomes evident that the international prognostic index (IPI), a standard for risk stratification, and COO subtyping do not identify patients who will not respond to and experience remission following the initial R-CHOP therapy [1,7,10]. Accordingly, it is valuable to determine predictive biomarkers to guide the rational use of R-CHOP therapy to optimize patients’ outcomes and avoid unnecessary treatment. The mechanisms underlying treatment resistance in DLBCL are multifactorial and include host variability, gene mutations and alteration of signal transduction pathways, tumor microenvironment, aberrant metabolic pathways, enhanced drug efflux via an overexpression of multidrug efflux transporters, and the upregulation of anti-apoptotic proteins [1,11,12,13].

The adenosine triphosphate binding cassette (ABC) transporters comprise a family of 48 transmembrane proteins which physiologically protect cells against xenobiotics, and when aberrantly expressed, play a role in the multidrug resistance (MDR) of tumors acting as a drug efflux from the cytoplasm domain [8,11,14]. Among the ABC transporters involved in MDR, the overexpression of P-glycoprotein (P-gp/ABCB1), multidrug resistance-associated protein 1 (MRP1/ABCC1), breast cancer resistance protein (BCRP/ABCG2), and subfamily C member 5 (ABCC5) had predictive values of treatment failure in patients with acute myeloid leukemia (AML), acute lymphocytic leukemia (ALL), Hodgkin lymphoma and DLBCL, respectively [15,16,17,18,19,20].

Among the important resistance mechanisms to immunochemotherapy in DLBLC is reduced susceptibility to apoptosis through synergistic effects of increased expression of Bcl-2, considered as a poor predictive indicator in R-CHOP-treated patients; gene alterations of Bcl-6, MYC and TP53; and the inhibitor of apoptosis protein (IAP) family [7,11]. Survivin, the most studied IAP, has a bifunctional role acting in mitosis regulation beyond apoptosis inhibition. It can be detected in different subcellular localizations such as the cytoplasm, nucleus and mitochondria, indicating different functions. Survivin impairs caspase 3, 7 and 9 activations by stabilizing XIAP (X-linked IAP) directly binding to caspases, enhancing the activity of other IAPs, regulating cell death by preventing APAF-1 (apoptotic protease activating factor-1) and releasing Smac/Diablo (second mitochondria-derived activator of caspase/direct inhibitor of apoptosis-binding protein) [21,22]. Several studies have implicated survivin in multidrug resistance of most human cancers including lymphoma, AML and chronic myeloid leukemia (CML) [22,23]. However, with regard to patients with de novo DLBCL treated with an R-CHOP regimen, in several studies, the prognostic value of survivin was confirmed, and conversely, other authors demonstrated that survivin expression was prognostically irrelevant [24,25]. 

Due to the non-clarity of their clinical relevance in comparison to Bcl-2, we investigated if coexpression of the most common MDR proteins (P-gp/ABCB1, MRP1/ABCC1, and BCRP/ABCG2) and survivin is associated with R-CHOP treatment resistance in previously untreated patients with DLBCL.

## 2. Materials and Methods

### 2.1. Patient Selection

This study included 73 consecutive patients with histopathologically diagnosed DLBCL, de novo (initial diagnosis) or at relapse where it was appropriate, and who were treated with R-CHOP or rituximab-containing regimen (equivalent immunochemotherapy) in our institution from 2015 to 2021. Inclusion criteria were available medical history, diagnostic paraffin blocks of lymph nodes, and planned first-line treatment with four to eight cycles of R-CHOP or equivalent treatment. The diagnosis was established according to the criteria of the World Health Organization classification [26]. The staging of the disease was done according to the Ann Arbor classification [27]. According to Ann Arbor criteria, patients with advanced-stage disease received radiotherapy to bulky sites after R-CHOP. Treatment response was assessed according to standard criteria [28,29,30]. Patients who achieved complete remission (CR) received a total of six to eight R-CHOP, while those in partial remission (PR) completed eight courses after four ones. The treatment response was evaluated at the end of the fourth course and the end of treatment. Follow-up visits were scheduled every 3 months for the first 2 years, and every 6 months thereafter. Patients achieving PR, refractory or relapsed patients (relapse at least 6 months and to 2 years after competition of R-CHOP) were treated with second-line (immuno-) chemotherapy regimens or autologous stem cell transplantation according to the local standard care of treatment. Refractory disease was defined in patients who achieved less than a partial response and had documented persistent disease after first-line therapy, or patients progressing within 3 months of completion of the first-line treatment. The patients were classified into immune types, either ABC or the GBC, according to the model proposed by Hans et al. [31]. 

Patients with transformation from low-grade B-cell lymphoma, primary central nervous system DLBCL, primary cutaneous DLBCL or AIDS/HIV-associated lymphoma were excluded.

Several clinical variables were particularly analyzed: age, gender, clinical stage, extranodal sites (Yes versus No) and corresponding number of the sites, performance status (PS) according to Eastern Cooperative Oncology Group (ECOG), B symptoms (Yes versus No), the Revised International Prognostic Index (R-IPI) score, ß2-microglobulin, and serum lactate dehydrogenase (LDH) (normal/decreased versus increased). The R-IPI score was determined as described previously [32]. 

### 2.2. Ethic Statement

This study complied with all the provisions of the Declaration of Helsinki and its current amendments and was conducted using the Good Clinical Practice Guidelines. The study was approved by the Ethics Committee of the Faculty of Medicine, University of Banja Luka (approval number 18/4.36/17).

### 2.3. Treatment

All patients were treated with the first line of immunochemotherapy: 63 patients received the R-CHOP regimen consisting of cyclophosphamide, 750 mg/m^2^; doxorubicin, 50 mg/m^2^; vincristine, 1.4 mg/m^2^ (up to a maximum dose of 2 mg) on day 2; and prednisone, 60 mg, administered orally, on days 2–6. Rituximab was administrated at a dose of 375 mg/m2 on day 1. The treatment was repeated every 3 weeks. Ten patients received the R-CHOEP (addition of etoposide, 50 mg/m on days 2–5) to R-CHOP. The patients in clinical stages II-IV were treated with six to eight cycles of R-CHOP or equivalent immunochemotherapy. The patients in the I clinical stage were treated with 3 cycles of immunochemotherapy and field radiotherapy. In those with “bulky” disease, radiotherapy was applied after immunochemotherapy.

### 2.4. Immunohistochemistry

The lymph node sections for routine hematoxylin and eosin (HE) staining from each patient with DLBCL were analyzed. All diagnostic biopsy specimens were analyzed using a conventional light microscopy examination and immunohistochemical (IHC) analysis, independently by two blinded pathologists, and disagreements were reanalyzed until they reached a consensus. The formalin-fixed, paraffin-embedded lymph node tissue slides underwent deparaffinization and heat-induced antigen retrieval techniques. 

Immunohistochemistry (IHC) studies with antibodies for Bcl-2 (clone 100/D5; dilution 1:100; Abcam, ab692), P Glycoprotein/ABCB1 (clone EPR10364-57; dilution 1:1200; Abcam, ab170904), survivin (clone EP2880Y, dilution 1:250, Abcam, ab76424), MRP1/ABCC1 (clone MRPm5, dilution 1:100, Abcam, ab24102) and BCRP/ABCG2 (clone BXP-21, dilution 1:75, Abcam, ab3380) were performed on 3–4 µm tissue sections using the streptavidin–biotin complex technique, according to the manufacturer’s instructions. The IHC identification of the investigated mediators was performed with the application of the UltraVision LP Detection system: HRP polymer & DAB Plus Chromogen (Product # TL-125-HD, Thermo Scientific™ Lab Vision™ UltraVision™, Fermont, CA, USA). The 3.3′-diaminobenzidine (DAB) (Thermo Fisher Scientific, Fermont, CA, USA) was used as a chromogenic substrate, and contrasting was performed with hematoxylin. The control sections were immunostained under identical conditions, substituting the primary antibody with a buffer solution. Stained specimens were analyzed at an objective magnification of ×200 investigators (light microscope Leica DM2500, Microsystems GmbH, Wetzlar, Germany).

The expression of certain molecules was quantified by calculating the percentage (%) of immunopositive cells (only cells three times larger than small lymphocytes with intensive optical density expression of Bcl-2, P-gp/ABCB1, survivin, MRP1/ABCC1 and BCRP/ABCG2 across ten non-successive fields using ImageJ software 1.50 (National Institute of Health, Bethesda, Rockville, MD, USA), and assessed semiquantitatively using immunoreactive score (IRS)). Survivin expression was considered immunopositive when lymphoma cells showed cytoplasmic and/or nuclear staining [33,34].

The intensity of membranous/cytoplasmic and/or nuclear staining was designated as: (1) negative or weak, (2) moderate or (3) strong. The percentage of immunopositive cells was scored as either no cells (0), ≤ 50% of cells (1), 51–75% of cells (2) or ≥ 76% of cells stained (3) [25,35]. Via the multiplication of these two parameters, IRS was calculated for each antigen and grouped as positive antigen expression (IRS ≥ 4.5) and low antigen expression (IRS < 4.5).

### 2.5. Statistical Analysis

The results were analyzed with the statistical software package SPSS Statistics for Windows v26 (IBM Corporation, Armonk, New York, NY, USA). Mann–Whitney’s U test was used for the comparison of continuous variables, whereas Pearson’s chi-square analysis or Fisher’s exact test was applied for the comparison of categorical variables. We conducted a univariate and multivariate binary logistic regression analysis to determine independent predictive factors for the DLBCL treatment resistance to the first-line treatment. *p* < 0.05 was considered statistically significant.

## 3. Results

### 3.1. Clinical Characteristics of the Patients

The patients’ main demographic and clinical characteristics according to their response to the first treatment (R-CHOP or equivalent) are summarized in Table 1.

The median age of patients in the study was 61 years (range, 19–83 years); 45 patients (61.6%) were men. Complete remission was achieved in 23 (31.5%) patients. After a median follow-up of 40 months, 30 (41.1%) patients relapsed at least 6 months after the treatment had been completed, whereas 20 (27.4%) had initially refractory disease. A total of 38 patients (52.1%) were classified as GBC subtypes. Univariate analysis showed that the following clinical parameters were significantly associated with the first-line treatment response: ECOG (*p* < 0.001), R-IPI (*p* < 0.05), ß2-microglobulin (*p* = 0.035), extranodal localization (*p* = 0.03), and clinical stage (*p* = 0.015). Multivariate analysis showed that only ECOG was an independent predictive factor for the R-CHOP treatment response (*p* < 0.001) in patients with DLBCL.

### 3.2. Bcl-2 Expression and Correlation with Clinicopathologic Variables

Increased Bcl-2 expression, as the indicator of poor response to R-CHOP, varied among the different patient groups. The results showed that the cytoplasmic expression of Bcl-2 was significantly increased at initial diagnosis in the Refractory and Relapsed groups (72.3% ± 12.3 and 64.1% ± 13.6 cells, respectively, *p* < 0.01 compared to the Remission group) (Figure 1). 

According to IRS, Bcl-2 expression was determined to be high (≥4.5) in both Relapsed (mean IRS 6.4 ± 1.88) and Refractory (mean IRS 8.42 ± 2.83) groups and low (<4.5) in the Remission group, respectively. The relationship between different clinicopathologic variables and Bcl-2 expression showed statistically significant correlations with either extranodal localization (*p* < 0.05), R-IPI (*p* < 0.05) or ECOG (*p* < 0.05) among the patients’ groups. 

### 3.3. ABC Transporter Expression in DLBCL

We tested the expression of three ABC transporters (P-gp/ABCB1, MRP1/ABCC1, and BRCP/ABCG2) in DLBCL specimens in all R-CHOP-or-equivalent-treated patients. Marked expression of P-gp/ABCB1 was detected in the Refractory (45.6% ± 10.3, *p* < 0.001 vs. Remission) group, while there was no difference at initial diagnosis in Remission and Relapsed DLBCL cells (30.8% ± 10.3 and 29.2% ± 9.1 cells, respectively, *p* > 0.05) (Figure 2). 

However, in semiquantitative analysis, there was no substantial difference in the IRS of P-gp/ABCB1 among the groups showing a low IRS (mean IRS 3.12 ± 1.05). 

MRP1/ABCC1 and BRCP/ABCG2 were markedly expressed by DLBCL cells in all patient groups. A significant high expression of MRP1/ABCC1 was detected at initial diagnosis in the Refractory and Relapsed DLBCL groups (70.33% ± 12.18, *p* = 0.002, and 56.86% ± 11.29, *p* = 0.03 vs. Remission, respectively). The staining reaction was localized in the cytoplasm membrane and selected cytoplasm granulations with different saturation in individual cases (Figure 2). According to the semiquantitative analysis of MRP1/ABCC1 expression, the mean IRSs were 7.7 ± 2.11 SD in the Relapsed group and 8.2 ± 1.47 in the Refractory group, *p* < 0.001, vs. a low IRS in the Remission group, respectively.

For BRCP/ABCG2, the staining localization was predominantly distributed to the cytoplasm membrane with different saturation in individual cases. The BRCP/ABCG2 expression level at initial diagnosis in the Refractory DLBCL was significantly higher (54.59% ± 8.27) than either Relapsed (44.81% ± 5.06) or Remission DLBSL (29.03% ± 8.6), *p* < 0.001, respectively (Figure 2). According to a semiquantitative analysis of BRCP/ABCG2 expression, the mean IRSs were 7.10 ± 1.81 SD in the Relapsed group and 7.7 ± 1.95 in the Refractory group, *p* < 0.001, vs. a low IRS in the Remission group, respectively.

Furthermore, potential correlations were tested between either Bcl-2 or among different ABC transporters in DLBCL. There was a moderate positive correlation between P-gp/ABCB1 and either survivin or MRP1/ABCC1, *p* < 0.001, respectively, in each DLBCL group. In contrast, there was no correlation between P-gp/ABCB1 and either Bcl-2 or BRCP/ABCG2 or clinicopathological data. We found strong positive correlations between the expressions of MRP1/ABCC1 and Bcl-2 (*p* < 0.001) (Figure 3a,d,g), and moderate ones with BRCP/ABCG2 (*p* < 0.05) (Figure 3c,f,i) in all the analyzed DLBCL groups. 

Furthermore, the calculations demonstrate that there was a strong positive correlation between BRCP/ABCG2 and Bcl-2 (*p* < 0.05) (Figure 3b,e,h) in all the analyzed DLBCL groups. 

Finally, strong correlations were found between MRP1/ABCC1 and either R-IPI (*p* < 0.05) or ECOG (*p* < 0.05) and a moderate correlation between BRCP/ABCG2 and ECOG (*p* < 0.05) in the Relapsed and Refractory groups, respectively. 

### 3.4. Survivin Expression and Correlation with Bcl-2 and ABC Transporters

Survivin showed significant immunoexpression at initial diagnosis either in Remission (39.4% ± 11.3), Relapsed (52.0% ± 13.9, *p* < 0.05 vs. Remission), or Refractory DLBCL cell (74.6% ± 15.3, *p* < 0.001 vs. Remission, *p* = 0.33 vs. Relapsed) groups (Figure 4).

A semiquantitative analysis of survivin expression showed a high IRS (mean IRS 6.67 ± 1.2) in the Refractory and Relapsed groups and a low IRS (mean IRS 3.88 ± 0.8) in the Remission group, respectively. As survivin is one of the IAPs, we tested the correlation of coexpression with Bcl-2 and each ABC transporter. As illustrated in Figure 5a–i, strong correlations were determined between survivin and either Bcl-2 (Figure 5a,d,g) or MRP1/ABCC1 expression (%) (Figure 5b,e,h), *p* < 0.001, respectively, and moderate correlations with either P-gp or BRCP/ABCG2 (*p* < 0.05) (Figure 5c,f,i) in all DLBCL groups. 

### 3.5. Subgroup Analysis

Paired tissue specimens of the same patient in the Relapsed group at initial diagnosis and relapse were analyzed for Bcl-2, P-gp/ABCB1, MRP1/ABCC1, BRCP/ABCG2, and survivin expression. The expression of Bcl-2 in the paired samples did not significantly change. Significantly increased expression of P-gp/ABCB1 (48.85% ± 14.20, *p* < 0.01 vs. initial diagnosis) was detected at relapse but remained at a low IRS. In all paired specimens in the Relapsed DLBCL group, slightly higher expression of MRP1/ABCC1 and BRCP/ABCG2 was detected at relapse (n.s vs. initial diagnosis). Only, survivin expression was significantly higher at relapse (*p* < 0.05 vs. initial diagnosis) (Figure 6).

### 3.6. Linear Regression Analysis to Predict R-CHOP Treatment Resistance

In the linear regression model, all clinical and molecular parameters (Bcl-2, P-gp, MRP1/ABCC1, BRCP/ABCG2, and survivin) were analyzed to determine the independent predictive factors for the DLBCL treatment resistance to the first-line treatment. Except for ECOG, no significant predictive value was found in the following clinicopathologic parameters: age, gender, Ann Arbor stage, B symptoms, R-IPI score, number of extranodal localizations, LDH, β2 microglobulin, and cell of origin. The results of the multiple linear regression analysis are presented in Table 2.

ECOG along with MRP1/ABCC1, BRCP/ABCG2, and survivin were associated with the R-CHOP treatment response (adjusted R^2^ = 0.63, *p* < 0.05). The higher the ECOG status (B = 0.412, *p* < 0.001) or the higher expression of MRP1/ABCC1 (B = 0.024, *p* < 0.001) or BRCP/ABCG2 (B = 0.033, *p* < 0.001), the higher the rate of R-CHOP failure or occurrence of refractory or relapsed disease, respectively. Considering the strong correlation between survivin and MRP1/ABCC1 (*p* < 0.001), survivin is considered a positive predictive factor for R-CHOP treatment resistance.

## 4. Discussion

In the study, we investigated the expression of survivin and three ABC transporters in patients with DLBCL and analyzed correlations between IHC expression and clinicopathological parameters and their potential predictive values to the R-CHOP treatment response. The main findings showed that the overexpression of MRP1/ABCC1, survivin and BRCP/ABCG2 was significantly associated with either relapsed or primary refractory DLBLC and it can be highly predictable for R-CHOP treatment failure. All DLBCL cases were positive for P-gp/ABCB1 expression, but with a low IRS and insignificant predictive value for R-CHOP response.

The anti-apoptotic protein Bcl-2 was discovered as the protooncogen involved in chromosomal translocation (t(14;18) (q32;q21)) in follicular lymphoma, and its overexpression is associated with drug resistance. In DLBCL, the frequency of Bcl-2-positive cases is highly variable, ranging from 24% to 80% in examined specimens in the previous studies that used IHC; so, its predictive relevance to the first-line treatment is considered controversial [36]. 

Consistently, our results showed highly increased expression in relapsed and refractory DLBCL but a wide range of Bcl-2 expression in lymphoma cells including the patients who achieved CR (remission). In addition, multivariate analysis did not confirm the predictive value of Bcl-2 to R-CHOP response. Today, double Hit (DH) or Double Expresser (DE) DLBCL comprises ∼20–25% of newly diagnosed high-grade (HG) DLBCL, is defined by either the translocation or overexpression of MYC and Bcl-2 (less frequently, Bcl-6), and is correlated with R-CHOP treatment resistance with a 2-year overall survival of ∼20% [37,38,39]. However, the predictive value of Bcl-2, Bcl-6, or MYC for treatment efficacy might be debatable [8,40,41]. Thus, drug resistance becomes an obstacle and an investigation of the mechanism underlying R-CHOP resistance may be beneficial to DLBCL treatment. 

Due to the heterogeneous genetic nature of DLBCL, multiple molecular mechanisms are required for the intrinsic resistance and acquisition of chemotherapy resistance in DLBCL [42]. Previous research demonstrated that the abnormal expression levels of ABC transporters were associated with poor clinical outcomes of various malignancies, including high-risk DLBCL, but their overexpression does not appear to be consistently associated with treatment resistance in DLBCL [16,42,43,44].

Doxorubicin, vincristine, and prednisone are substrates for P-gp/ABCB1 and are known to induce *MDR-1* expression [8]. Contrary to our expectation, the low immunoreactivity score (IRS) of P-gp/ABCB1 was detected in all DLBCL specimens but not considered a predictive factor for R-CHOP response. Furthermore, significantly higher expression at relapse in comparison to the initial diagnosis indicates an upregulation of P-gp/ABCB1 after R-CHOP and a potential role in induced treatment resistance. Although there have been several studies, conflicting results still exist on the predictive value of P-gp/ABCB1 for response to R-CHOP or equivalent treatment [44,45].

Expression of the multiple ABC transporters in most cases of lymphoma may suggest a role in the development of drug resistance. Steidl et al. [46] showed an overexpression of MRP1/ABCC1 in the therapy-resistant classical Hodgkin-lymphoma-derived (CHL) cell line KMH2. Similarly, Greaves et al. [17] detected the significant expression of MRP1/ABCC1 and BRCP/ABCG2 in CHL specimens. Patients with MRP1/ABCC1 expression had a higher risk of treatment failure and no correlation with other clinical parameters. However, BRCP/ABCG2 was not significantly associated with the treatment refractoriness of CHL. Our results, similar to the study by Gündüz et al. [47] and Oshawa et al. [16], showed an overexpression of MRP1/ABCC1 and a high IRS at initial diagnosis, particularly in patients who would develop R/R DLBCL after first-line R-CHOP treatment, so that it may serve as a predictive indicator to the treatment response. In addition, the important finding is that MRP1/ABCC1 had a significant positive correlation with antiapoptotic markers, survivin and Bcl-2 in R/R DLBCL.

We also showed an increased expression of BRCP/ABCG2 in R/R DLBCL, with good predictive value to R-CHOP response but a strong correlation only with Bcl-2. These results indicate that BRCP/ABCG2 may play a role in DLBCL R-CHOP resistance. Similar to our IHC results, Singh et al. [15] showed the coexpression of Bcl-2 and BRCP/ABCG2 as an underlying mechanism of DLBCL chemoresistance. Furthermore, they demonstrated that the upregulation or inhibition of the hedgehog (Hh) signaling pathway, a key factor behind the expression of BRCP/ABCG2, may present a therapeutic target in overcoming chemoresistance in DLBCL.

Earlier studies have shown conflicting results about the predictive relevance of survivin for the treatment response and associated survival time in patients with DLBCL treated with R-CHOP [25,48]. However, the significant improvement in DLBCL patients’ survival following the addition of rituximab to CHOP warranted a re-evaluation of such predictive factors. In a large multicentric study that included patients with de novo DLBCL who were treated with R-CHOP, it was confirmed that survivin was an independent predictive factor for poor outcomes in ABC-DLBCL, and the patients seemed to benefit less from this treatment and may require additional novel agents [26]. Accordingly, a meta-analysis provided robust evidence on the prognostic and clinicopathological value of the increased expression of survivin in DLBC in terms of poor prognosis, clinical stage, and treatment response [24].

Our results showed the coexpression of survivin and Bcl-2 positively correlated in the relapsed group and also in primary refractory DLBCL, which indicates an upregulation of anti-apoptotic and pro-survival mechanisms in the lymphoma cells. In addition, the coexpression of survivin and MRP1/ABCC1 indicates a potential drug-resistant mechanism through the activation of drug efflux pumps as ABC transporters (MRP1/ABCC1 as well as BRCP/ABCG2) and inhibition of lymphoma cell apoptosis. Similarly, the coexpression of survivin and P-gp/ABCB1 was determined to be responsible for resistance to the conventional treatment of Burkitt lymphoma (BL), showing that treatment with verapamil, a P-gp/ABCB1 inhibitor, or Survivin siRNA suppressed the proliferation of all drug-resistant BL cells [45]. 

It has been confirmed that CHOP chemoresistance is mainly associated with the upregulation of anti-apoptotic and pro-survival (Bcl-2 family, Bcl-XL, BCR signaling, and IAPs) signaling pathways in DLBCL [11,42].

Besides the complement-dependent cytotoxicity and antibody-dependent cellular toxicity, rituximab’s anticancer effects involve the activation of apoptosis by inhibiting pro-survival pathways such as PI3K/AKT/mTOR (phosphatidylinositol-3-kinase (PI3K))/Akt and the mammalian target of rapamycin (mTOR), PI3K/AKT/NF-kB (nuclear factor kappa B), p38 MAPK (*p38* mitogen-activated protein kinase), and MEK/ERK (mitogen-activated protein kinase/extracellular signal-regulated kinase) [11]. One of the underlying mechanisms in tumor resistance is the aberrant activation of PI3K/AKT pathways that upregulates the expression of ABC transporters (P-gp/ABCB1, MRP1/ABCC1, and BRCP/ABCG2), and pro-survival molecules such as Bcl-2 and IAPs [11,49]. 

Therefore, dysfunctional PI3K/AKT pathways may serve as a hub to synergistically activate the coexpression of IAPs, Bcl-2, and R-CHOP drug efflux through the ABC transporter [49].

Finally, it has been proposed that rituximab resistance is mostly related to the downregulation of CD20 expression or *MS4A1(CD20)* gene mutations in de novo and R/R disease, but its clinical relevance is not yet fully understood [11,42]. 

## 5. Conclusions

In summary, the overexpression of ABC transporters (MRP1/ABCC1, BRCP/ABCG2, and IAP) and survivin at the de novo diagnosis of DLBCL is associated with relapsed and primary refractory disease and an increased risk of R-CHOP treatment resistance. Although it remains to be elucidated in clinical settings, the coexpression of either MRP1/ABCC1 and survivin with/without BRCP/ABCG2 and other clinicopathological markers is likely to improve the predictability of treatment resistance. Additionally, MRP1/ABCC1, survivin, and BRCP/ABCG2 may serve as potential targets for therapeutic applications of inhibitors in increasing susceptibility to immunochemotherapy.

## Figures and Tables

**Figure 1 cancers-15-04106-f001:**
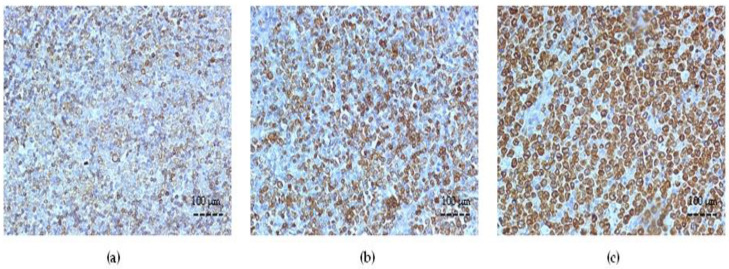
Representative examples of immunohistochemical staining for Bcl-2 in DLBCL (×200 magnification). (**a**) Remission, low IRS; (**b**) Relapsed, high IRS (≥4.5); (**c**) Refractory, high IRS (≥4.5). Abbreviations: DLBCL—diffuse large B-cell lymphoma; IRS—immunoreactive score.

**Figure 2 cancers-15-04106-f002:**
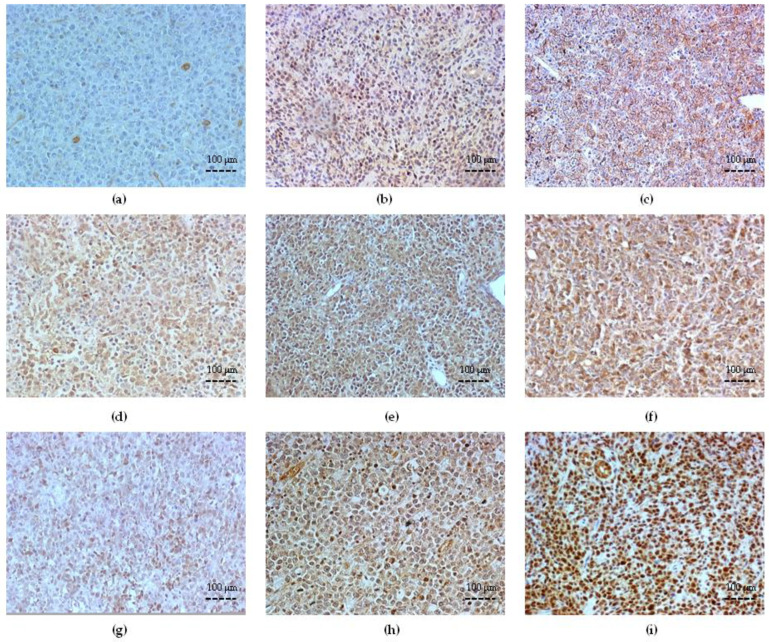
Immunohistochemical staining for ABC transporter expression in DLBCL (×200 magnification). (**a**–**c**) Representative examples of positive expression for P glycoprotein/ABCB1 (P-gp) with low IRS. (**a**) Remission; (**b**) Relapsed; (**c**) Refractory; (**d**–**f**) representative examples of positive expression for MRP1/ABCC1; (**d**) Remission, low IRS; (**e**) Relapsed, high IRS (≥4.5); (**f**) Refractory, high IRS (≥4.5). (**g**–**i**) Representative examples of positive expression for BCRP/ABCG2. (**g**) Remission, low IRS; (**h**) Relapsed, high IRS (≥4.5); (**i**) Refractory, high IRS (≥4.5). Abbreviations: ABC—adenosine triphosphate binding cassette; DLBCL—diffuse large B-cell lymphoma; IRS—immunoreactive score.

**Figure 3 cancers-15-04106-f003:**
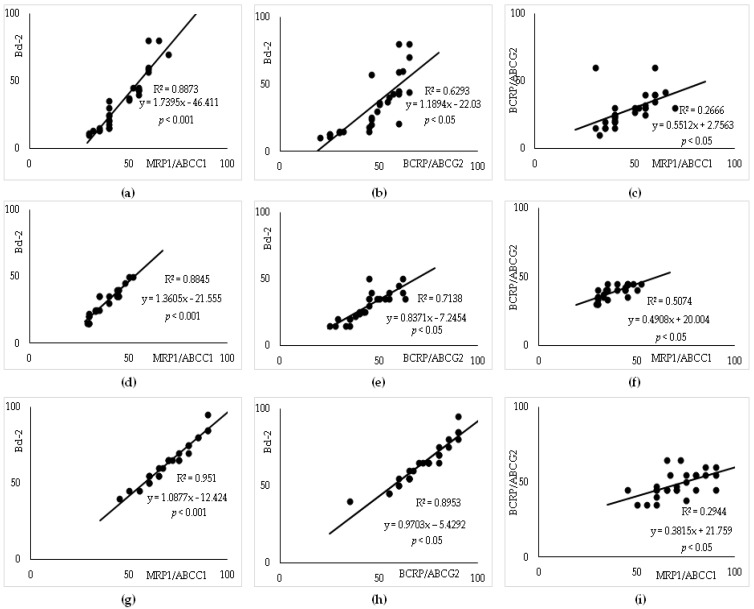
Strong positive correlations among ABC transporters and Bcl-2 expression (%) in DLBCL. The correlations among Bcl-2, MRP1/ABCC1, or BCRP expression (%). (**a**–**c**) Remission; (**d**–**f**) Relapsed; (**g**–**i**) Refractory DLBCL. Abbreviations: ABC—adenosine triphosphate binding cassette; DLBCL—diffuse large B-cell lymphoma.

**Figure 4 cancers-15-04106-f004:**
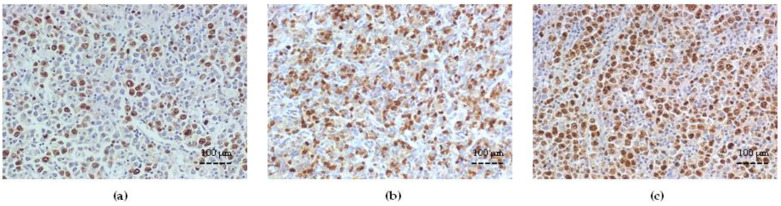
Representative examples of immunohistochemical staining for survivin in DLBCL (×200 magnification). (**a**) Remission, low IRS; (**b**) Relapsed, high IRS (≥ 4.5); (**c**) Refractory, high IRS (≥ 4.5). Abbreviations: DLBCL—diffuse large B-cell lymphoma; IRS—immunoreactive score.

**Figure 5 cancers-15-04106-f005:**
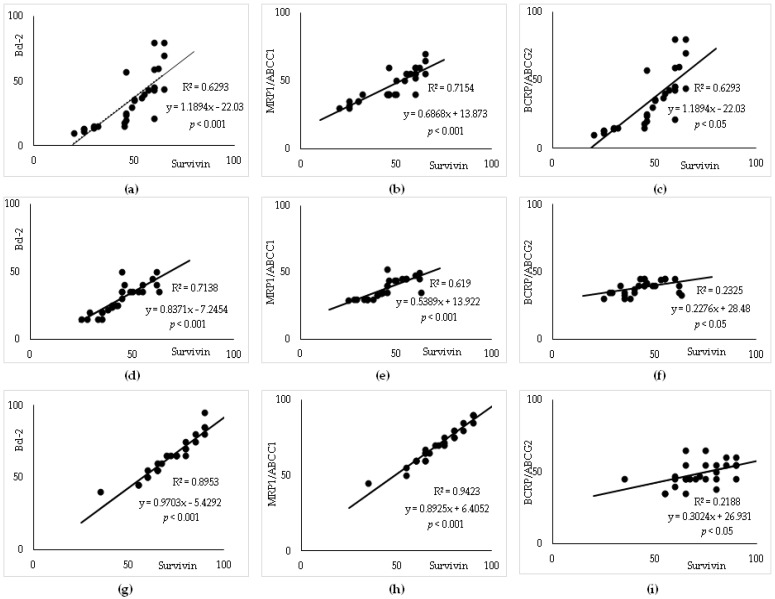
Significant correlations between survivin, ABC transporters and Bcl-2 expression (%) in DLBCL. (**a**–**c**) Remission; (**d**–**f**) Relapsed; (**g**–**i**) Refractory DLBCL. Abbreviations: ABC—adenosine triphosphate binding cassette; DLBCL—diffuse large B-cell lymphoma.

**Figure 6 cancers-15-04106-f006:**
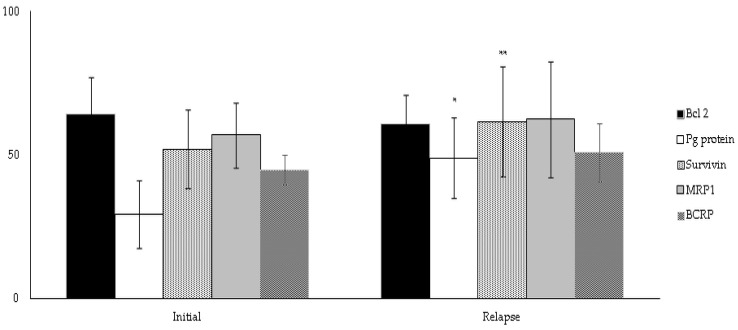
Comparison of expression (%) of Bcl-2, P-gp/ABCB1, MRP1/ABCC1, BRCP/ABCG2, and survivin in paired samples in Relapsed DLBCL at initial diagnosis and relapse. * *p* < 0.05, ** *p* < 0.01 vs. initial diagnosis). Abbreviations: DLBCL—diffuse large B-cell lymphoma.

**Table 1 cancers-15-04106-t001:** Clinical data and histological characteristics of all patients with diffuse large B-cell lymphoma.

Characteristics	Patients with DLBCL, n = 73 (%)
Age (median)	61
range	19–83
≥ 60 years	(46.6)
Gender	
Male/female	45 (61.6)/28 (38.4)
Treatment response	
Complete remission	23 (31.5)
Relapsed disease	30 (41.1)
Primary refractory disease	20 (27.4)
Stage	
I	4 (5.5)
II	4 (5.5)
III	14 (19.2)
IV	51 (69.8)
PS (ECOG) > 1	38 (52.1)
B symptoms	47 (64.4)
R-IPI score	
Low (0)	1 (1.4)
Intermediate (1–2)	35 (47.9)
High (>2)	37 (50.7)
Extranodal localization > 1	18 (24.7)
LDH elevated	46 (63.0)
β2 microglobulin	31 (41.5)
Treatment	
R-CHOP/ Equivalent R-CHOP	63 (86.3)/10 (13.7)
GBC/non-GBC subtype	38 (52.1)/35(47.9)

Abbreviations: DLBCL—diffuse large B-cell lymphoma; GBC—germinal center B-cell-like subtype DLBCL; LDH—lactate dehydrogenase; PS—performance status; ECOG—Eastern Oncology Cooperative Group; R-IPI score—Revised International Prognostic Index; R-CHOP—cyclophosphamide, doxorubicin, vincristine, prednisone.

**Table 2 cancers-15-04106-t002:** Results of multiple linear regression analysis to predict diffuse large B-cell lymphoma response to R-CHOP or equivalent first-line treatment.

Model	Unstandardized Coefficients		Standardized Coefficients		*p*
	B	S.E.	Beta	t	
All Patients with DLBCL (n = 73)					
Constant	−0.774	0.309		−2.505	0.014
ECOG	0.412	0.102	0.316	4.062	<0.001
MRP1/ABCC1	0.024	0.006	0.327	3.907	<0.001
Survivin	0.027	0.006	0.332	3.988	<0.001
BCRP/ABCG2	0.033	0.007	0.374	4.740	<0.001

Abbreviations: DLBCL—diffuse large B-cell lymphoma; R-CHOP—cyclophosphamide, doxorubicin, vincristine, prednisone; ECOG—Eastern Cooperative Oncology Group; PS—performance status; S.E.—standard error.

## Data Availability

Data availability is limited due to institutional data protection laws and the confidentiality of patient data.

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
