# Peer review of "Overexpression of MRP1/ABCC1, Survivin and BCRP/ABCC2 Predicts the Resistance of Diffuse Large B-Cell Lymphoma to R-CHOP Treatment"

_cancers, 2023, doi:10.3390/cancers15164106_

Round 1
Reviewer 1 Report
This is an interesting manuscript aiming to identify if ATP binding cassettes transporters overexpression may correlate to residence to the standard therapy (R-CHOP) in patients with DLCBL. While the investigation appears solid, I would suggest some revisions to increase the value of the author’s work.
· Kaplan-Meier estimates of overall survival and progression-free survival for the relapsed and refractory DLBCL patients vs. patients in remission stratified by level of expression of the investigated biomarkers should be added.
· The Correlations graphs should include the individual p values for each correlation in the corresponding figure.
· All correlations mentioned in the results section must be shown as figure where applies.
· Were biopsy specimens in the refractory group collected (beside the ones at the time of diagnosis?
Minor comment:
· The shadow beside each figure should be removed. They are not recommended for publication.
Moderate editing of English language required
Author Response
Response to Reviewer 1 Comments
Point 1: This is an interesting manuscript aiming to identify if ATP binding cassette transporters overexpression may correlate to residence to the standard therapy (R-CHOP) in patients with DLCBL. While the investigation appears solid, I would suggest some revisions to increase the value of the author’s work.
Kaplan-Meier estimates of overall survival and progression-free survival for the relapsed and refractory DLBCL patients vs. patients in remission stratified by level of expression of the investigated biomarkers should be added.
Response 1: Thank you so much for your general comments and useful suggestions.
Dear reviewer, yes indeed, we agree with your observation. The main objective of our study was to explore predictive factors (ABC transporters, Bcl-2, and survivin) of DLBCL to the first-line treatment response. The patients in remission (who had a complete response) were regularly assessed over the proposed 2 years after the first-line treatment was completed, and there were no lethal cases. The patients who were diagnosed with relapsed (at least 6 months to 2 years after the first line treatment had been completed) or refractory disease (for the criteria please see section 2.1, page 3, lines 152-154), and according to the local standard of care procedures, they underwent to the next assessments (clinical, laboratory, staging), and as soon as possible continued with the second line treatments (chemotherapy or immunochemotherapy or transplantation) depends on the disease stage, and clinical status. Therefore, for the purpose of this study, we could not follow and analyzed overall survival (OS) after R-CHOP failure in these patients, and the Kaplan-Meier curve was not shown. As the patients were stratified according to the treatment response and known criteria, progression-free survival (PFS) was not analyzed by the expression of each biomarker. Instead, multiple linear regression including ABC transporters, survivin and ECOG (clinical value) was done to determine predictive factors for the DLBCL R-CHOP treatment response (Table 2, Page 10). Based on this finding, further studies will examine OS and PFS in patients with CR after R-CHOP or equivalent, stratified by the level of expression of the certain biomarker.
Point 2: The Correlations graphs should include the individual p values for each correlation in the corresponding figure.
Response 2: We included the individual p values for each correlation in the corresponding figure.
Please see:
Section: 3.3 Page 8, Figure 3;
Section 3.4 Page 9, Figure 5.
Point 3: All correlations mentioned in the results section must be shown as a figure where applies.
Response 3: Thank you. We added a corresponding Figure mark in the Results where applicable.
Please see:
Section: 3.3 Page 8, Lines 291-293 and Lines 297, 298, 301 and 305.
Section 3.4, Page 9, Lines 320-323, and Lines 326-327.
Point 4: Were biopsy specimens in the refractory group collected (besides the ones at the time of diagnosis?
Response 4: For this study, we did not repeat the biopsy once we confirmed the refractory disease in a patient. According to the standard of care procedures, these patients underwent other assessments (staging, laboratory finding etc.) and then switched to other treatment options. Please see Section 2.1. Patient selection for the definition of Refractory disease. Page 3, Lines 152-154.
Point 5: Minor comment: The shadow beside each figure should be removed. They are not recommended for publication.
Response 5: The shadows beside each figure were removed.
Please see Page 6, Figure 1; Page 7, Figure 2; Page 8, Figure 3; Page 9, Figure 4, Page 9, Figure 5, Page 10, Figure 6.
Point 6: Comments on the Quality of the English Language Moderate editing of the English language is required.
Response 6: Thank you very much for these remarks. The whole manuscript has been revised. Also, proofreading, grammar and linguistic improvements were performed by a court interpreter for the English language, given by the Medical Faculty of the Military Medical Academy, University of Defence in Belgrade, Serbia.
Please see the attachment.

Reviewer 2 Report
In the present study, the authors investigated the correlation between ABC transporters and survivin expression in the context of R-CHOP treatment response in DLBCL. The study is novel and of clinical importance in the treatment of DLBCL. I have a few minor concerns as shown below:
1. Is there any direct regulation between ABC transporters and survivin expression in DLBCL? The authors could knock down either of the molecules and analyze the expression of the other in DLBCL cell lines.
2. Did the authors analyze other IAPs, than survivin? Is the phenomenon universal for all the IAPs or only specific for survivin?
3. The manuscript should be thoroughly proof-read to eliminate the typos and grammatical errors. For example, Abstract, line 61: Add 'the' before 'Aim was....'; line 74: It should be DLBCL instead of DCLB.
4. Add the term 'Abbreviations' before describing the full forms of the terms of the tables.
5. The authors should discuss how R-CHOP treatment influences the expression of ABC transporters and survivin and vice versa.
Author Response
Response to Reviewer 2 Comments
Point 1: In the present study, the authors investigated the correlation between ABC transporters and survivin expression in the context of R-CHOP treatment response in DLBCL. The study is novel and of clinical importance in the treatment of DLBCL. I have a few minor concerns as shown below:
Is there any direct regulation between ABC transporters and survivin expression in DLBCL? The authors could knock down either of the molecules and analyze the expression of the other in DLBCL cell lines.
Response 1: Under answer 5, we discussed a link between ABC transporters and survivin. To our best knowledge, this is one of the recent explanations related to human oncology including haematological malignancies in clinical settings as well as in cell lines. However, direct regulation among ABC transporters and survivin has not been reported yet.
Please see answer 5.
Point 2: Did the authors analyze other IAPs, than survivin? Is the phenomenon universal for all the IAPs or only specific for survivin?
Response 2: For this study, only survivin expression, as the most studied IAP, as well as co-expression with ABC transporters and Bcl-2, were analyzed due to the following reasons: a) confirmed involvement of survivin in multidrug resistance of human hemato-oncology [please see Introduction, lines 117-125, reference 22,23]; b) conflicting results about independent predictive values of survivin in relapsed and refractory DLBCL [please see Introduction, lines 125-128, reference 24,25]; c) of most importance was to determine the predictive significance of co-expression of survivin with ABC transporters and Bcl-2 in R-CHOP resistant DLBCL, and d) survivin is recognized as a universal target antigen for the developing antitumor immunotherapy.
Point 3: The manuscript should be thoroughly proofread to eliminate typos and grammatical errors. For example, Abstract, line 61: Add 'the' before 'Aim was....'; line 74: It should be DLBCL instead of DCLB.
Response 3: The whole manuscript has been revised and we eliminated the typos and grammatical errors. Besides, proofreading, grammar and linguistic improvements were performed by a court interpreter for the English language, given by the Medical Faculty of the Military Medical Academy, University of Defence in Belgrade, Serbia.
- We added 'the' before 'Aim was....';.
Please see Page 2, Line 8.
- Also we replaced DCLB with DLBCL.
Please see Page 2, Lines 3 and 21.
Point 4: Add the term 'Abbreviations' before describing the full forms of the terms of the tables.
Response 4: Thank you. Please see the correction in terms of Abbreviations for each Table and Figure.
Please see Table 1, Page 6; Figure 1, Page 6, lines 247-249; Figure 2, Page 7, lines 262-269; Figure 3, Page 8, lines 296-299; Figure 4, Page 9, lines 314-316; Figure 5, Page 10, lines 326-327; Figure 6, Page 10, lines 340; Table 2, Page 10, line 349.
Point 5: The authors should discuss how R-CHOP treatment influences the expression of ABC transporters and survivin and vice versa.
Response 5: Thank you for your suggestion.
This is explored in the Discussion citing the main signalling pathways involved in the R-CHOP mechanism of action, mainly of rituximab, elucidating its additional known anticancer effect through inhibition of pro-survival and antiapoptotic molecules (Bcl-2 and IAPs such as survivin), and ABC transporters expression. The main signalling pathway PI3K/AKT is explored in terms of the R-CHOP mechanism of IAPs and ABC regulation. Reference number 50 is added, accordingly.
Please see Discussion, page 23, lines 441-452, and reference 50, page 15, lines 620-621.
Round 2
Reviewer 1 Report
I would like to thank the authors for their pertinent responses to the reviewer's questions. I believe that the manuscript can be published in the present form. Thank you.